# Exploring Sex-Based Neuropsychological Outcomes in Pediatric Brain Cancer Survivors: A Pilot Study

**DOI:** 10.3390/diseases12110289

**Published:** 2024-11-11

**Authors:** Chiara Colliva, Veronica Rivi, Pierfrancesco Sarti, Isabel Cobelli, Johanna M. C. Blom

**Affiliations:** 1Local Health Unit of Modena, District of Carpi, 41012 Carpi, Italy; chiara.colliva@unimore.it; 2Department of Biomedical, Metabolic and Neural Sciences, University of Modena and Reggio Emilia, 41125 Modena, Italy; veronica.rivi@unimore.it (V.R.); pierfrancesco.sarti@unimore.it (P.S.); 298441@studenti.unimore.it (I.C.); 3Department of Adult Psychiatry and Psychotherapy, Psychiatric Hospital, University of Zurich, 8008 Zurich, Switzerland; 4Center for Neuroscience and Neurotechnology, University of Modena and Reggio Emilia, 41125 Modena, Italy

**Keywords:** pediatric oncology, executive function, chemotherapy, cisplatin, carboplatin, cognitive impairment, gender medicine, neuropsychological assessment, resilience

## Abstract

This pilot study investigated sex differences in neuropsychological outcomes between pediatric brain cancer patients treated with cisplatin and/or carboplatin. Using structured neuropsychological assessments, we evaluated the executive functions in 17 pediatric cancer survivors. The BRIEF and TOL tests measured key executive function domains, while the Resiliency Scale and PAT test assessed resilience and family psychosocial risk. Our results revealed significant sex differences, with the males having higher scores than the females in inhibitory control, impulse regulation, and strategic planning. These findings underscore the complexity of cognitive outcomes in pediatric cancer survivors and highlight the importance of understanding sex-specific differences for developing tailored interventions.

## 1. Introduction

Over the past decade, there has been a remarkable increase in the number of tumor survivors, with projections indicating a substantial rise in the coming years [1,2]. According to the Italian Association of Medical Oncology, from 2007 to 2019, the recorded number of deaths caused by tumors in both men and women was consistently lower than expected based on the average rates from 2003 to 2006 [3]. Additionally, the European Cancer Information System reports that the 5-year survival rate for children with cancer has improved significantly, with an overall survival rate of approximately 80% across Europe [4]. The American Cancer Society also estimates that the population of cancer survivors in the United States will soar to 20.3 million by 1 January 2026, marking a significant milestone in survivorship [5]. This positive trajectory is further supported by encouraging statistics revealing a continued decline in cancer mortality rates since 2021, surpassing the milestone of averting over 4 million fatalities since 1991 [6]. This trend spans all age groups, including children and adolescents, as evidenced by epidemiological data showing a notable incidence rate of 155.8 per million person-years among those aged 0–19 years [2]. However, increased survival rates are often accompanied by a higher susceptibility to various somatic, cognitive, psychological, and psychiatric sequelae [7,8]. These include depression; posttraumatic stress disorder; sleep disturbances; and significant socioeconomic consequences, such as prolonged unemployment and disrupted educational trajectories [9,10]. Many cancer patients experience cognitive impairments commonly referred to as “chemo brain”, “chemo fog”, or “cancer-related cognitive impairment” [11,12]. These impairments affect key cognitive domains, including memory consolidation, attentional control, problem-solving, and cognitive flexibility, and are observed in both active patients and those in remission [13,14,15,16]. Such cognitive deficits can persist for many years after treatment, posing a significant challenge for children and adolescents, whose developmental stages are marked by ongoing neuroplasticity and cognitive refinement [17,18]. These challenges profoundly affect the well-being and quality of life of survivors and their caregivers, often persisting for many years beyond the initial diagnosis [17,19].

The complex experience of pediatric cancer patients is fraught with challenges, as the diagnosis itself serves as a traumatic event that triggers psychological distress for both the children and those around them [13]. Hospitalization and the associated stressors can further exacerbate these effects, creating a complex landscape of mental and emotional health concerns [20].

Moreover, the treatments can introduce additional adverse outcomes, impacting the developing brain during crucial growth periods [21]. Among the drugs used in oncology settings, cisplatin is a cornerstone drug used as a first-line therapy against many solid tumors, including ovarian, head and neck, testicular, and small-cell lung cancers, due to its potent anticancer activity [22,23,24,25]. It is often administered as an adjuvant treatment following surgery and radiation for solid tumors [17,22]. Several analogs of cisplatin, such as carboplatin, are also widely used in clinical practice. The anticancer efficacy of cisplatin is primarily attributed to its interaction with DNA, which disrupts replication and inhibits tumor growth [23,24]. Despite its therapeutic benefits, cisplatin is associated with severe toxic effects, notably neurotoxicity and nephrotoxicity [25]. These side effects highlight the need for ongoing research to develop less toxic cisplatin analogs and combination therapies that can mitigate these adverse effects [17,23]. Beyond the cognitive domain, cisplatin’s impact extends to sensory modalities, manifesting in visual and auditory disturbances [23,25]. Additionally, chemotherapeutic agents, like cisplatin, have been implicated in disrupting neurotransmitter systems, particularly dopamine-mediated pathways critical for prefrontal cortex–limbic circuitry, which may underlie the observed cognitive sequelae [26,27,28,29].

In this complex scenario, there is increasing evidence that sex differences significantly influence cancer prevention, susceptibility, progression, survival, treatment responses, and post-treatment consequences [30]. This emerging understanding underscores the importance of considering biological sex as a key factor in the personalization of cancer treatment and long-term survivorship care. Recognizing these differences is essential for tailoring therapies that not only target the cancer effectively but also mitigate long-term cognitive and psychosocial impacts specific to each sex. By integrating sex-based considerations into treatment and care planning, clinicians can offer more individualized interventions that enhance both the efficacy of treatment and the overall quality of life for survivors and their families.

Sex differences in executive functions among children and adolescents without cancer were observed in a range of cognitive domains, and these differences are thought to emerge due to a combination of biological, developmental, and environmental factors [31,32,33,34]. Research consistently showed that males and females may exhibit distinct strengths and weaknesses in executive functions [35,36,37]. For instance, females tend to outperform males in tasks involving impulse control, sustained attention, and verbal fluency, which are critical components of executive function. Boys, on the other hand, may demonstrate better performance in visuospatial tasks and problem-solving strategies that require abstract thinking and cognitive flexibility [37]. These differences are particularly notable during adolescence, a developmental period marked by the significant maturation of brain structures related to executive functions, such as the prefrontal cortex [38,39,40]. Hormonal changes associated with puberty, particularly the effects of estrogen and testosterone on brain development, are believed to play a key role in shaping these cognitive differences [41]. Social and environmental factors, such as gendered expectations and educational practices, may also contribute to the observed disparities between boys and girls in executive functioning [39].

Despite the substantial body of literature highlighting cognitive impairments associated with cancer treatments, there remains a gap in understanding how these effects differ by sex. Specifically, previous studies largely focused on the overall cognitive impact of treatments, without investigating potential sex differences that could inform tailored interventions and support mechanisms for affected individuals [9,15]. Thus, understanding these sex differences in neurotypical children and adolescents provides a valuable context for assessing how cancer treatments, such as those involving cisplatin and carboplatin, may differentially affect executive functions in male and female pediatric cancer survivors. This underscores the importance of tailoring interventions to address the specific cognitive needs of each sex in both clinical and educational settings.

In light of these concerns, this study aimed to investigate sex differences in neuropsychological outcomes, specifically focusing on executive functions and psychosocial factors in pediatric and adolescent patients treated for solid tumors with cisplatin and/or carboplatin. Given the potential impacts of both the cancer diagnosis and treatment on cognitive development, this pilot study sought to provide preliminary insights into whether these treatments affected males and females differently in terms of executive functioning and psychosocial health.

The primary aim of our pilot study was to investigate these sex differences in cognitive functions between cancer patients treated with cisplatin and carboplatin. By elucidating whether the cognitive outcomes varied by sex, our research aimed to identify specific vulnerabilities and resilience factors that may influence cognitive health in this population. This approach can not only enhance our understanding of the cognitive sequelae of cancer treatments but also addresses a critical gap in the literature regarding sex-based differences in treatment effects.

To achieve this, we employed a range of neuropsychological assessments, including the Psychosocial Assessment Tool (PAT) to identify families with high psychosocial risk [42], the Resiliency Scale for Children and Adolescents to assess vulnerability and protective factors [41], and the Behavior Rating Inventory of Executive Function (BRIEF) to evaluate executive function in daily life [43]. Additionally, problem-solving ability was assessed using the Tower of London (TOL) test [44,45].

## 2. Materials and Methods

### 2.1. Participants

This pilot study recruited a cohort of pediatric and adolescent cancer survivors from the pediatric oncology department of the university hospital “Azienda Ospedaliero-Universitaria Policlinico di Modena” (Italy). Before enrollment, written informed consent was obtained. For younger children and adolescents who were unable to provide informed consent due to their age, their parents provided full consent on their behalf. In cases where participants were adolescents with the cognitive capacity to understand this study, they were also informed about this research and provided their assent, alongside the consent from their parents or legal guardians. This process ensured that the rights and welfare of all participants, particularly minors, were appropriately safeguarded throughout this study. In any case, although some patients were over 18 years old, all were accompanied by their family members.

The cohort comprised 17 individuals, consisting of 11 females (64.71%) and 6 males (35.29%), with ages that ranged from 2 months to 22 years and a mean age of 11.26 ± 7.3 years. Importantly, this was a representative sample of the region where the selection took place, consistent with data from Italy and Europe [45,46,47,48]. All participants had been diagnosed in the past 10 years with pediatric solid brain tumors, where the age of diagnosis was 6.9 ± 7.3 years. The mean duration of the disease was 10.29 months, with a standard deviation of ±9.84 months.

Importantly, at the time of the current evaluation, all the participants and their families exhibited sufficient proficiency in the Italian language. Language proficiency was evaluated using standardized assessments specifically designed for various developmental stages. This meticulous approach ensured that all participants possessed the necessary communication abilities for meaningful engagement in this study, which eliminated any ambiguity regarding their language proficiency and enhanced the overall integrity of this research. Moreover, it is important to note that the parents were consistently present for all participants, including those who were of legal age. The impact of the condition was such that they did not attend evaluations independently. Both the BRIEF assessments for adult participants and those completed by parents were comparable in terms of standardization and scoring. The Parent-Reported Outcomes (PROs) were not differentiated, as participants completed them in conjunction with their parents. However, the participants completed the TOL task independently. The Resiliency Scale assessment was also conducted individually.

### 2.2. Treatments

All individuals in the sample had undergone chemotherapy treatment, where the treatment regimens varied between participants: 4 individuals received the Cisplatin treatment (23.53%), 9 individuals received the Carboplatin treatment (52.94%), and 4 individuals were treated with a combination of Cisplatin and Carboplatin (23.53%). The treatment had been completed before the commencement of this study and its duration ranged from 1.5 months to 1.5 years. The dosages of the treatment were recorded, which considered factors such as the type and stage of cancer, risk group, and whether the treatment was initiated for an initial diagnosis or a relapse.

### 2.3. Intensity Treatment Rating Scale

The intensity of treatment for pediatric tumors was stratified using the Intensity of Treatment Rating Scale (ITRS), is a robust measure comprising intensity levels and content items [49]. The data from our cancer patients show ITR-3 classifications of 11.76% at level 2, 64.71% at level 3, and 23.53% at level 4, providing insights into the intensity of treatment that was administered and its implications for patient outcomes.

This scale allowed us to categorize pediatric cancer treatments based on their intensity, which ranged from level 1 (minimally intensive) to level 4 (most intensive). Having 11.76% of patients at level 2 in our data indicates that only a small portion underwent minimally intensive treatment. This low percentage suggests that most patients require more aggressive therapies, which is common in pediatric oncology due to the nature of the cancers treated in this population. The significant majority at level 3, which comprised 64.71%, reflects that these patients received moderately intensive treatment regimens, which likely included standard chemotherapy protocols and possibly radiation. This aligned with typical treatment pathways, indicating a balance between treatment efficacy and manageable side effects. The 23.53% of patients at level 4 signified that a notable segment received highly intensive treatments, such as bone marrow transplantation or aggressive chemotherapy for high-risk conditions. This necessitated close monitoring and supportive care, as these treatments came with increased risks of toxicity and long-term effects. The distribution of ITR levels emphasized the diversity in treatment approaches within our patient population. With many patients at level 3 and a significant proportion at level 4, it was necessary to consider the implications for both short-term and long-term management. Patients receiving higher-intensity treatments will likely require more intensive monitoring for potential side effects, both acute and chronic. Understanding these potential impacts, such as neurodevelopmental issues or psychosocial challenges, is essential for comprehensive patient care.

### 2.4. Patients’ Enrollment

Eligible patients and their parents were approached during routine follow-up audiometric assessments and invited to participate in this study. On the same day, written informed consent and participant anamnesis were obtained. In a designated setting, either before or after the audiometric assessment, participants completed the neuropsychological tests as part of the study protocol, while parents filled out the parent-reported questionnaires. All procedures were conducted by trained personnel. This approach facilitated the seamless integration of data collection into the existing clinical workflow, which ensured minimal disruption to the participants’ schedules while maximizing the efficiency of the data acquisition.

### 2.5. Tests, Questionnaires, and Measures

Several assessments were utilized to comprehensively evaluate executive functions (EFs) and psychosocial factors in pediatric and adolescent cancer survivors:Psychosocial Assessment Tool—2.0 (PAT 2.0): A parent-reported screening questionnaire assessing psychosocial risk in families of children with chronic illnesses [42,50].Behavior Rating Inventory of Executive Function (BRIEF): Questionnaire assessing EF behaviors in daily life, completed by parents [43,51].Tower of London (TOL) test: Neurocognitive test assessing planning, monitoring, and problem-solving abilities [44].Resiliency Scale for Children and Adolescents (RSCA): Questionnaire evaluating resilience factors [41].

The neuropsychological tests used in this study were widely validated and demonstrated high reliability across different populations, including pediatric cancer survivors. The BRIEF, for instance, is one of the most commonly used tools for assessing executive functions in children and adolescents. It showed excellent internal consistency, with Cronbach’s alpha values that typically ranged between 0.80 and 0.98 across its scales, and test–retest reliability coefficients that ranged from 0.82 to 0.88. The BRIEF was calibrated in various cultural contexts, including the Italian population, where it demonstrated similar reliability and validity values, which ensured its applicability in this study. The TOL test, employed to assess problem-solving and planning abilities, was extensively validated for both clinical and non-clinical populations. It exhibited a reliability coefficient that ranged from 0.74 to 0.87 and was shown to have strong construct validity in measuring executive function domains related to strategic thinking and goal-directed behavior. Italian normative data for the TOL also support its use in pediatric populations, ensuring that it was suitable for the age group and cultural context in this study. Additionally, the RSCA, used to evaluate vulnerability and protective factors, showed robust psychometric properties, with reliability scores for its subscales (sense of mastery, sense of relatedness, and emotional reactivity) that ranged between 0.80 and 0.90. The RSCA was also adapted and validated for use with Italian children and adolescents, which ensured its relevance to this specific cohort. These high reliability and validity values across all the neuropsychological instruments used in this study, together with the availability of Italian normative data, reinforced the robustness of the findings and the appropriateness of the selected tests for evaluating cognitive and psychosocial outcomes in this pediatric population.

### 2.6. Statistical Analysis

Prior to running the statistical tests, we performed several checks to verify these assumptions. For the parametric analyses, we assessed the normality of the data distributions using the Shapiro–Wilk test and inspected the skewness and kurtosis values. The homogeneity of variances was evaluated through Levene’s test to ensure that the group comparisons were appropriate. In cases where the parametric assumptions were not met, non-parametric alternatives were employed. These steps ensured that the analyses were conducted with robust and appropriate methods and provided reliable results despite the small sample size. Comparisons between the males’ and females’ scores were analyzed using unpaired *t*-tests. The participant performance was evaluated against the normative data of healthy subjects using appropriate statistical methods. Descriptive statistics (mean, median, standard deviation) were calculated to analyze the demographic variables. All statistical analyses were performed using GraphPad Prism v. 9.00e.

Network analysis was conducted using the mgm, qgraph, networktools, and botnet libraries. The variables were categorized into groups for better visualization. A Mixed Graphical Model (MGM) was fitted to the data using leave-one-out cross-validation—a special case of k-fold cross-validation where k equals the number of data points. Optimal regularization parameters were selected using the qgraph library, which also visualized the network by showing connections and variable groupings. The node predictability was assessed to measure how well each node was predicted by the rest of the network. Centrality measures, including strength, betweenness, closeness, and expected influence, were computed to determine the importance of each node in the network. A moderation analysis was conducted by including gender (male and female) as a categorical moderator within the network. This approach allowed for group comparisons within a single model framework, making it possible to assess the differences across multiple groups for all parameters simultaneously. This method is especially advantageous because it can be applied to all commonly used cross-sectional network models [52]. A continuous risk score was created for each patient by calculating a weighted sum of the relevant variables, with each variable assigned an equal weight of 0.1. The variables included in this calculation were age; sex; cisplatin (a chemotherapy drug); carboplatin (a chemotherapy drug); treatment duration; Psychosocial Assessment Tool (PAT) score; all Behavior Rating Inventory of Executive Function (BRIEF) subscales, excluding Behavioral Regulation Index (BRI), Metacognition Index (MI), and Global Executive Composite (GEC); violations; decision time; execution time; accuracy of the Tower of London test (a neuropsychological test); and mastery (MAS), relatedness (REL), resourcefulness (REA), and vulnerability (VULN) of the Resiliency Scale. This resulted in a composite risk score that combined the influence of all these variables. Patients were then classified as “High Risk” or “Low Risk” based on whether their risk score was above or below the median score. The “Risk” variable was converted into a factor to facilitate the categorical analysis. The dataset was then split into training and test sets using a 70/30 ratio, which ensured a random but reproducible partition by setting a seed. A Random Forest model (i.e., a machine learning algorithm) was trained on the training set to classify patients into high- and low-risk categories using the aforementioned variables as predictors. The importance of each variable in predicting risk was assessed. The model’s performance was evaluated on the test set using a confusion matrix, which provided metrics such as accuracy, sensitivity, and specificity. Finally, a statistical comparison was conducted between the low-risk and high-risk patient groups to identify significant differences between the variables that contributed to the risk score. For each pair of variables, the Wilcoxon rank-sum test (a non-parametric statistical test) was used.

## 3. Results

### 3.1. Descriptive Analyses

In this study, we enrolled a total of 17 participants. The mean age at cancer diagnosis was 6.9 years, with a median age of 3 years. The participants were diagnosed with solid tumors during childhood, which reflected the typical demographic profile of pediatric cancer cases [53]. The gender distribution in our sample was 64.71% female and 35.29% male. Regarding the age at diagnosis, 64.71% of participants were diagnosed at between 0 and 5 years of age, 11.76% between 6 and 13 years, and 23.53% between 14 and 22 years. This age distribution aligned with the expected onset for childhood cancers and emphasized this study’s focus on early-life diagnoses and considering all developmental ages. The participants received different chemotherapy protocols: 23.53% underwent Cisplatin-based therapy, 52.94% received Carboplatin, and 23.53% received a combination of Cisplatin and Carboplatin. These treatments are standard in pediatric oncology and target various cancers commonly found in children and adolescents. The stratification based on the Intensity of Treatment Rating Scale (ITRS-3.0) revealed that 11.76% of participants were classified as level 2 survivors (indicating lower-intensity treatments), 64.71% as level 3 survivors, and 23.53% as level 4 survivors (indicating higher-intensity treatments). This distribution provided insights into the treatment regimens and their impacts on the health outcomes of the study cohort. The ITRS-3.0 categorization provides insight into the intensity of the treatments received. The predominance of level 3 survivors indicates that the participants underwent moderate- to high-intensity treatments, which could be a critical factor that influenced their health outcomes and cognitive recovery.

### 3.2. Sex Differences in Psychosocial Risk

Psychosocial risk was assessed using the Psychosocial Assessment Tool (PAT) scores, which were analyzed separately for females and males. Statistical analysis using an unpaired *t*-test showed no significant difference in the PAT scores between the females and males (t = 0.93, df = 14, *p* = 0.36) (Figure 1A). The mean PAT score for the females was 0.65, while for the males, it was 0.83, with medians of 0.635 and 0.85, respectively. These scores indicate an overall moderate psychosocial risk across the cohort, with some variability between the genders. Mapping the PAT scores onto the Public Health Pyramid Model (PPHM) framework revealed that 80% of the female participants fell into the universal population category, suggesting relatively lower psychosocial risk with sufficient supportive resources. The remaining 20% were categorized as part of the targeted group, indicating some acute distress and specific risk factors (Figure 1B). Among the males, 50% were classified as universal and 50% as targeted, indicating a more diverse psychosocial risk profile compared with the females (Figure 1C). This gender difference may reflect the underlying differences in coping mechanisms or support systems available to females versus males.

### 3.3. Sex Differences in Executive Functions

Executive functions were assessed using BRIEF scores, including individual domains and broader indexes, such as the Behavioral Regulation Index (BRI) and the Metacognition Index (MI) (Figure 2). Unpaired *t*-tests revealed significant differences between the females and males in several domains: inhibit (t = 2.23, df = 12, *p* = 0.04), initiate (t = 3.714, df = 11, *p* = 0.0034), MI (t = 2.579, df = 12, *p* = 0.024), and plan/organize (t = 2.58, df = 12, *p* = 0.02). The females generally scored lower in these domains compared with the males, indicating potential differences in inhibitory control, task initiation, and planning abilities. When comparing the mean BRIEF scores of the patients to normative values, both the females and males generally reported scores similar to the control population. The males tended to score slightly higher in all the subscales, meaning that they had lower performances in all the variables measured but not an actual impairment. Overall, these differences emphasized the need for gender-specific studies and interventions to address executive function challenges in pediatric cancer survivors.

### 3.4. Sex Differences in Planning and Problem-Solving Abilities

The Tower of London (TOL) test was used to evaluate planning and problem-solving abilities (Figure 3). The unpaired *t*-tests revealed no significant differences between the females and males in the following measures: decision time (t = 0.2930, df = 10, *p* = 0.7755), execution time (t = 0.3579, df = 10, *p* = 0.7279), total time (t = 0.3713, df = 10, *p* = 0.7181), violation (t = 0.2722, df = 10, *p* = 0.7910), and accuracy (t = 0.3262, df = 10, *p* = 0.7510). These results suggest that the performances in the planning and executing tasks were comparable between the sexes within the study cohort. A comparison of the mean TOL scores between the patients and the control population indicated that the participants performed similarly to the normative samples across all assessed variables. This suggests that cognitive abilities related to planning and problem-solving were generally preserved despite undergoing cancer treatments.

### 3.5. Sex Differences in Resilience

Scores on the mastery (MAS), resilience (REL), reactivity (REA), and vulnerability (VUL) subscales of the Resiliency Scale were analyzed separately for the females and males (Figure 4). The unpaired *t*-tests showed no significant differences between the genders in the mastery (MAS) (t = 0.91, df = 13, *p* = 0.38), resilience (REL) (t = 0.19, df = 13, *p* = 0.84), reactivity (REA) (t = 0.6997, df = 13, *p* = 0.4965), or vulnerability (VUL) (t = 0.13, df = 13, *p* = 0.89) scores. This indicates that the levels of resilience and vulnerability were comparable between the genders within the study cohort. The categorization of MAS, REL, and REA scores based on the resilience levels revealed variability across participants. Most MAS, REL, and REA scores fell within the average range, indicating moderate levels of resilience and adaptive capacities among the participants. The vulnerability index showed that the majority of participants were classified as having a slightly above average level of vulnerability, suggesting that both the males and females exhibited similar levels of psychological adjustment following their treatments.

### 3.6. Network Analysis and Risk Profile

The exploratory network analysis (Figure 5) resulted in a structure of 19 nodes and 21 edges with a density index of 0.12. Node 1 (Cisplatin) was the only one with no connection to the others. The arrangement of the variables and the centrality analyses made the nodes on the resilience scale stand out as the most important in the hub function, and hence, being most often on the shortest paths among the others. The nodes with a betweenness greater than and equal to 1 were 18 (REA) and 11 (monitor), which refer respectively to the ability of the given patients to exert control over their emotions and the ability to monitor situations. Closely linked to these are the sense of mastery (MAS, node 16) and the PAT score (node 4), which represents the psychosocial risk factor. Node 4 (PAT) remained tied more to the Tower of London nodes. Of particular interest was the negative relationship between vulnerability (node 19) and treatment duration (node 3). The more the duration increased, the more the vulnerability score decreased. Table 1 lists the values of the centrality measures for each node.

The moderated Gaussian Graphical Model (GGM) analysis revealed two key differences between the networks of the females and males. First, within the female network, a strong negative connection was observed between vulnerability (VULN) and treatment duration (treatment_duration), indicating that the longer treatment periods were associated with a decrease in vulnerability among the females (Figure 6). This relationship appeared to be less pronounced in the male network. Second, in the male network, the child’s social and family environment, as assessed by the PAT, played a crucial role in mitigating vulnerability. This finding is consistent with previous research, which suggests that males tend to benefit more from external resources, such as social and familial support, whereas these factors are often less significant for females. This distinction underscores the importance of considering gender-specific dynamics when evaluating the impact of treatment and support systems on vulnerability in pediatric populations.

Network analysis facilitates the identification of variables that are more relevant to a treatment setting due to their connectivity and centrality. However, it is equally important to identify individuals at the highest risk and determine whether they share common factors that contribute to this risk, especially within the context of a small sample size. The resulting risk model demonstrated strong sensitivity, where it successfully identified all high-risk cases. However, it exhibited lower specificity, where it accurately identified only 50% of the low-risk cases. With a balanced accuracy of 75%, the model reflected a reasonable overall performance. In terms of predictive values, the Positive Predictive Value (PPV) was a moderate 66.67%, indicating that a significant portion of the high-risk predictions were accurate. In contrast, the Negative Predictive Value (NPV) was perfect at 100%, illustrating the flawless identification of low-risk cases. Nevertheless, the statistical significance of these findings was less compelling. The *p*-value for accuracy relative to the no-information rate suggests that the model’s accuracy did not significantly exceed what could be expected from random guessing. Furthermore, McNemar’s test indicates no significant difference in misclassification rates between the two classes, while the kappa analysis revealed moderate agreement beyond chance. Thus, while the model was proficient at detecting high-risk cases, its performance in identifying low-risk cases was less robust, and the overall accuracy did not significantly surpass random guessing when considering the *p*-values and confidence intervals.

The Wilcoxon rank-sum test indicated that only a limited number of variables demonstrated significant differences between the high- and low-risk groups, as most variables did not show statistically significant distinctions. Specifically, Figure 7 highlights the three significant variables of the risk score: PAT (W statistic = 46.0, *p* = 0.0267), initiate (W statistic = 45.5, *p* = 0.0290), and plan_org (W statistic = 50.0, *p* = 0.0075).

## 4. Discussion

This pivotal study explored the impact of platin-based chemotherapy on the executive functions of male and female pediatric cancer survivors.

We enrolled 17 pediatric cancer survivors, with a mean age of 6.9 years and a median age of 3 years at diagnosis. This demographic aligned with the typical profile of pediatric cancer patients, who are diagnosed with solid tumors at a young age [46,47]. Notably, 64.71% of the participants were diagnosed between the ages of 0 and 5 years, reflecting the high incidence of early childhood cancers [54]. Additionally, the sex distribution in our study was representative of the usual sex ratio observed in pediatric oncology [39], which allowed us to consider our sample as a microcosm of the broader pediatric cancer population.

To validate our sample size, we reviewed existing studies focused on the neuropsychological and functional outcomes of pediatric brain tumor survivors and related conditions. Many of these studies are similarly constrained by small sample sizes, which can impact the generalizability of their findings. For example, Panwala et al. (2019) [55] analyzed 45 adult survivors of pediatric posterior fossa brain tumors, a small cohort reflective of the rarity of this diagnosis. Erickson et al. (2019) [56] reported on 79 participants, including 26 pediatric cancer survivors, highlighting the difficulties in recruiting sufficient numbers within such specific populations. Gandy et al. (2022) [57] examined 408 patients over ten years, yet data collection challenges over extended periods, including loss to follow-up, limited the robustness of their findings. Furthermore, Puhr et al. (2021) examined 48 eligible participants out of 90 in their study on executive function and psychosocial adjustment in adolescent survivors of pediatric brain tumors [31]. These examples underscore the necessity for multi-center collaborations to enhance sample sizes, allowing for more comprehensive analyses of long-term effects in pediatric cancer survivors.

To ensure the validity of our findings, we employed a comprehensive series of neuropsychological assessments to evaluate various domains of executive functions, including inhibitory control, planning, organizing, metacognition, and emotional regulation. These assessments were conducted using standardized procedures by trained professionals to minimize measurement errors and ensure consistency.

In contrast with previous research that has often reported significant cognitive impairments associated with cisplatin treatment [25,27,58], we did not observe a substantial impact on executive functions within our cohort. Our findings instead suggest that carboplatin might have a more pronounced effect on decision-making and task execution related to executive functions.

Although these treatments did not result in significant impairments in overall cognitive abilities, there were noteworthy differences in the executive functions between the sexes. This finding suggests that the cognitive impact of these chemotherapy drugs may not be uniform across all patients but could vary based on sex. The observation of sex differences in executive functions was a critical insight from our study. Executive functions, which include skills such as planning, problem-solving, and inhibitory control, are essential for adaptive behavior and everyday functioning [31].

Here, we found that the male and female survivors exhibited different patterns of executive function performance, which highlights the need for sex-specific considerations in survivorship care.

These differences may be attributed to various factors, including biological, hormonal, and psychosocial influences. For instance, sex hormones were shown to affect brain function and cognition, potentially leading to differential impacts of chemotherapy on males and females [36,57]. Additionally, psychosocial factors, such as stress responses and coping mechanisms, which can vary between sexes, may also play a role in shaping cognitive outcomes [48].

The identification of sex-based differences in cognitive outcomes underscores the importance of developing tailored interventions for cancer survivors.

Current survivorship care often adopts a one-size-fits-all approach, which may not adequately address the unique needs of each individual. By acknowledging and addressing sex differences, healthcare providers can design more personalized strategies that target specific cognitive and psychosocial challenges [59].

For example, intervention programs could be adjusted to focus on the particular cognitive deficits observed in male and female survivors. This might involve targeted cognitive rehabilitation therapies, stress management programs, and psychosocial support tailored to the specific needs of each sex [60]. Additionally, educational and vocational support could be customized to better align with the cognitive strengths and weaknesses identified in this study.

Moreover, the network analysis revealed that resilience, particularly in its components of sensitivity, recovery, and emotional reactivity, played a central role in the interaction framework between various variables.

Resilience, as conceptualized in our study, involves the ability to transform negative emotions through externalization [41,61]. This process is significant in pediatric patients, where caregivers and specialists play a critical role in managing these emotions [62,63,64]. Failure to externalize or communicate emotions can strongly influence other variables, with high centrality values indicating that resilience components are integral to the overall network affecting cognitive outcomes.

The risk model developed in our study also highlighted the importance of the psychosocial environment, in addition to frontal aspects of planning and initiative. The PAT emerged as one of the three most influential variables in this model, underscoring the need to monitor the social and family environment in pediatric oncology. These factors can significantly impact the likelihood of adverse events and cognitive outcomes.

The findings from this study have broader implications for cancer survivorship research and clinical practice, highlighting the necessity of incorporating sex-based considerations into both research and care practices [39,65]. Understanding how sex differences interact with cancer treatment and survivorship outcomes can lead to more nuanced and effective interventions.

Despite these promising results, the small sample size remained a significant limitation that potentially reduced the statistical power needed to detect subtle cognitive differences attributable to the cisplatin treatment.

However, we would like to remark that the challenges of recruiting participants with this particular diagnosis are well documented, particularly within the context of pediatric oncology. Patients diagnosed with solid tumors, especially in children and adolescents, present barriers to recruitment, such as limited prevalence rates and the sensitive nature of their medical conditions. The rarity of certain solid tumors means that potential participant pools are inherently small, resulting in a reduced ability to recruit a larger sample size. Our sample size reflected these constraints, which highlighted the difficulty of conducting research in this specialized area. Despite the limited number of participants, it is essential to note that our findings still contribute valuable insights into the cognitive and psychosocial outcomes of young cancer survivors. The small but focused sample allowed for an in-depth exploration of the specific nuances related to sex differences in executive functions and psychosocial factors, which may be overlooked in broader studies with more heterogeneous populations.

Moreover, our study’s pilot nature underscores the importance of exploring these dimensions, paving the way for larger, more comprehensive studies in the future. This foundational research can help to inform future investigations, which may ultimately lead to improved understanding and tailored interventions for this vulnerable population. By addressing these recruitment challenges head-on, we hope to advance this field and contribute to the growing body of literature on pediatric cancer survivorship.

The lack of baseline assessments before chemotherapy and the absence of a control group further limited our ability to make conclusive statements about the direct effects of cisplatin on cognitive functions.

Future research should also address these limitations by incorporating larger samples and considering the family environment as a potential protective factor against adverse outcomes.

Research indicated that caregivers and family dynamics evolve in response to a child’s illness, creating new needs and structures that must be understood and managed [63]. Resilience remains a crucial factor in coping with the traumatic experience of cancer treatment. Thus, therapies focusing on externalizing negative emotions, enhancing initiative, planning, and self-recovery are essential for helping children adjust better to their condition.

Given the observed sex-specific differences in executive functions within our cohort, it is also important to investigate the influence of sex hormones, genetic predispositions, and psychosocial factors on cognitive outcomes following cancer treatment. This approach would support the development of personalized interventions tailored to mitigate cognitive deficits and promote adaptive functioning in pediatric cancer survivors.

Moving forward, future research should build upon our preliminary findings by expanding the sample size to include a larger cohort of pediatric cancer survivors treated with cisplatin, which would enhance the statistical power and generalizability of results. Longitudinal studies are crucial for capturing the long-term trajectory of cognitive outcomes, incorporating repeated assessments to provide a nuanced understanding of chemotherapy’s impact on cognitive development over time.

By addressing the limitations of our study and pursuing these research opportunities, we can advance our understanding of cognitive outcomes in pediatric cancer survivors. Integrating interdisciplinary approaches and collaborative efforts will be essential in translating research findings into clinical practice, ultimately improving the quality of survivorship care. Prioritizing preventive strategies, such as early screening, lifestyle interventions, and supportive care measures, can help mitigate long-term toxicities, reduce the disease burden, and enhance the overall quality of life for pediatric oncology patients and survivors.

## 5. Conclusions

In conclusion, our study contributes to the evolving field of pediatric oncology by highlighting the complexities of balancing treatment efficacy with long-term cognitive outcomes. First, it provided new insights into the neuropsychological outcomes of pediatric and adolescent cancer survivors treated with cisplatin and carboplatin, particularly focusing on the underexplored area of sex differences in executive function. While previous studies examined the cognitive impact of cancer treatments, few systematically addressed the distinct cognitive profiles of male and female survivors in this population, especially in terms of inhibitory control, cognitive flexibility, and problem-solving abilities. Second, our study highlighted the critical role of sex as a variable that should be considered when developing personalized interventions for cancer survivors. By identifying notable sex differences in executive functions, our findings suggest that treatment plans and post-treatment cognitive rehabilitation should be tailored to address these specific needs, which have been largely overlooked in the literature to date. Additionally, our work incorporated validated neuropsychological assessments, such as the BRIEF and Tower of London tests, which provided reliable and robust data on executive function performance. The inclusion of the Resiliency Scale and PAT test added a psychosocial dimension to the analysis, emphasizing the interplay between cognitive function and family dynamics, which has not been comprehensively explored in similar studies. Finally, as a pilot study, our research laid the groundwork for future investigations by offering a preliminary model of how sex-specific factors influence cognitive outcomes after cancer treatment, which can guide larger longitudinal studies aimed at refining these findings and informing clinical practice. This study not only raises awareness of the cognitive and psychosocial sequelae of pediatric cancer treatment but also advocates for personalized, sex-based interventions to improve survivorship outcomes.

By embracing multidisciplinary approaches, addressing sex-specific differences, and leveraging innovative research methodologies, we can enhance our understanding and management of cognitive impairments in pediatric cancer survivors. Ultimately, prioritizing survivorship care that minimizes long-term toxicities while maximizing quality of life underscores the ethical imperative of pediatric oncology to ensure holistic patient care and well-being.

## Figures and Tables

**Figure 1 diseases-12-00289-f001:**
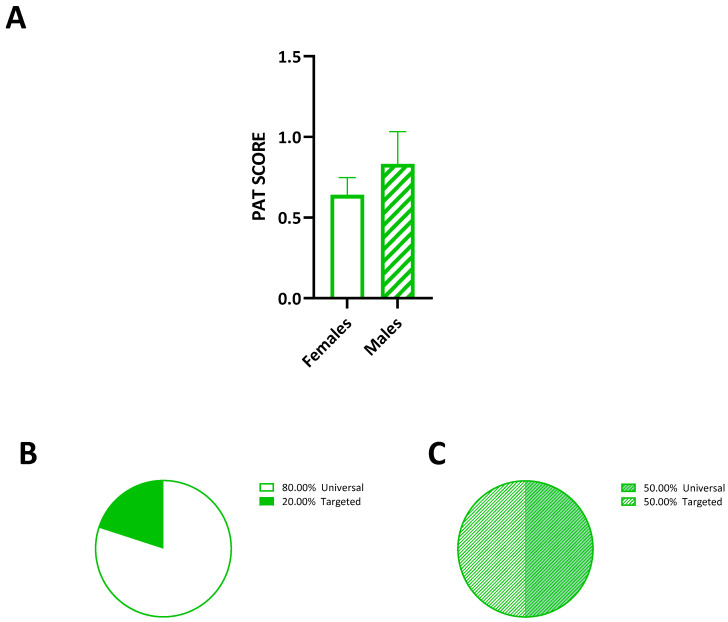
(**A**) PAT scores in males and females. (**B**) A total of 80.00% were categorized as part of the universal group, while 20.00% were classified as part of the targeted one. (**C**) The pie chart designates male categorization, showing that 50.00% were part of the universal population and 50.00% were part of the targeted one. Comparisons were made by an unpaired *t*-test. The thicker line is the mean, and the error bars are the s.e.m.s.

**Figure 2 diseases-12-00289-f002:**
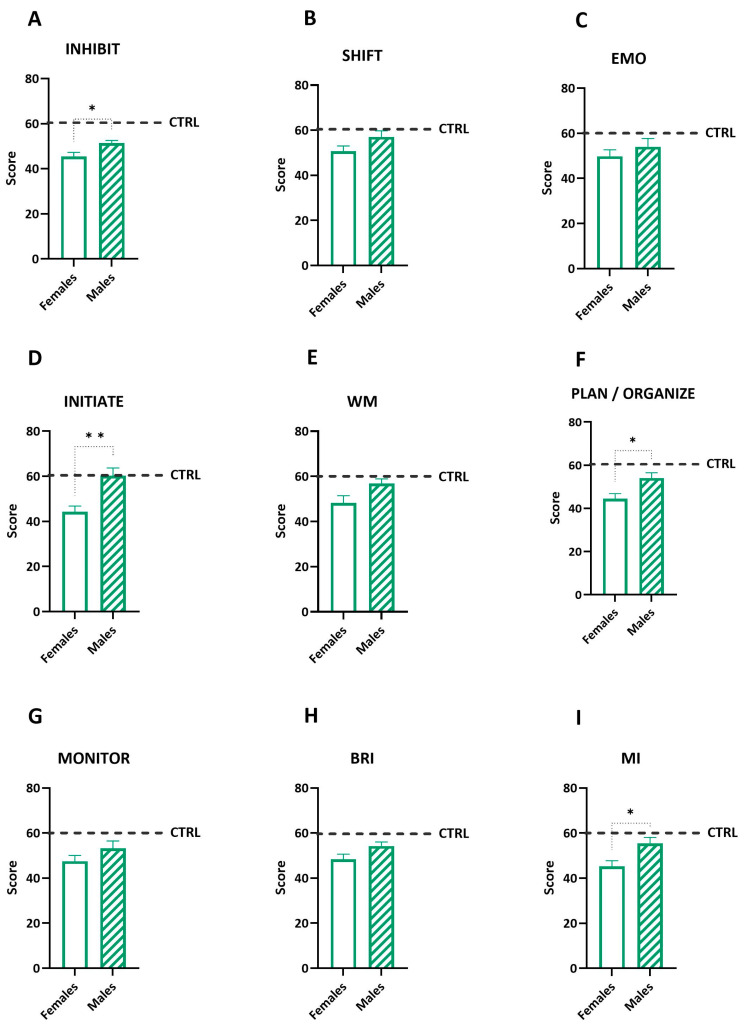
BRIEF variables’ scores were measured in the sample separately for the females (white bar) and males (diagonal bars). Significant differences between the females and males in the BRIEF inhibit (**A**), initiate (**D**), and plan/organize (**F**) scores emerged. However, no significant differences were found in the BRIEF shift, emotional control (EMO—(**C**)), working memory (WM—(**D**)), and monitor (**G**) scores. For the broader indexes, the BRI scores (**H**) showed no significant difference between the females and males, while the MI scores (**I**) revealed a significant difference between the sexes. No effects were found in the (**B**) SHIF and (**E**) WM, Comparisons were made by an unpaired *t*-test. The thicker line is the mean, and the error bars are the s.e.m.s., ** *p* < 0.01, * *p* < 0.05. The normative scores are represented by the dashed line.

**Figure 3 diseases-12-00289-f003:**
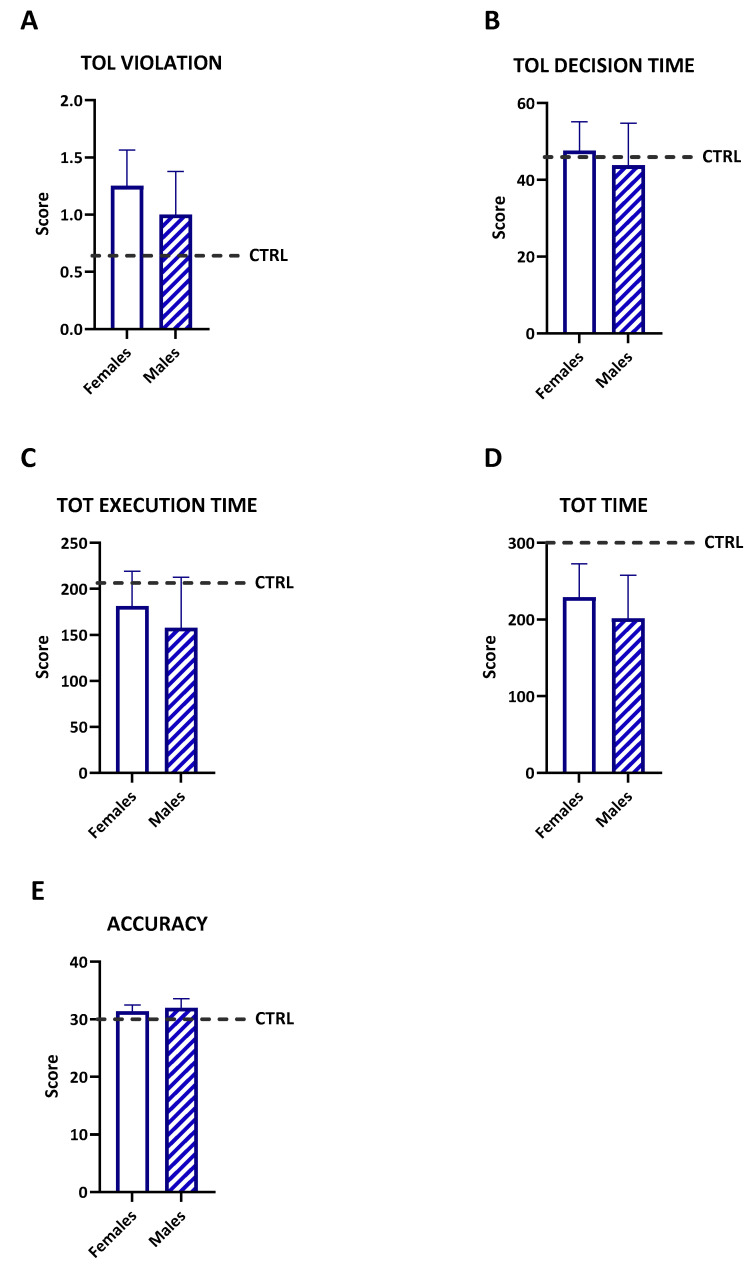
The TOL variables’ scores (**A**–**E**) TOL violation, TOL decision time, TOT execution time, TOT time, ACCURACY) were measured in the sample separately for the females (white bar) and males (diagonal bars). No significant differences between the females and males emerged. Comparisons were made by an unpaired *t*-test. The thicker line is the mean, and the error bars are the s.e.m.s. The normative scores are represented by the dashed line.

**Figure 4 diseases-12-00289-f004:**
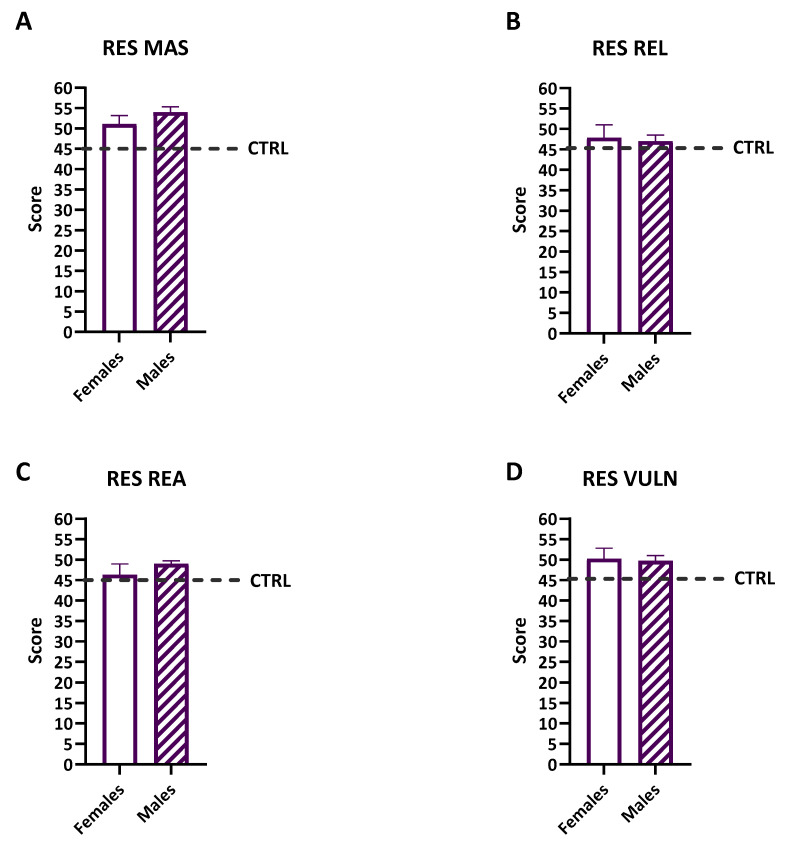
The females’ (white bar) and males’ (diagonal bars) scales in the different resiliency subscales: the sense of mastery scale (MAS—(**A**)), the sense of relatedness scale (REL—(**B**)), the emotional reactivity scale (REA—(**C**)), and the general vulnerability index (VUL—(**D**)). No significant differences between the females and males emerged. The comparisons were made by an unpaired *t*-test. The thicker dashed line is the mean, and the error bars are the s.e.m.s. The normative scores are represented by the dashed line.

**Figure 5 diseases-12-00289-f005:**
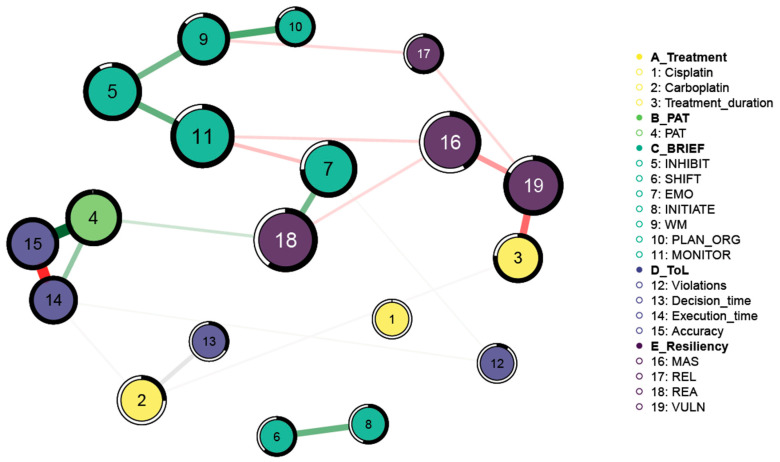
Representation of the network model of the 17 patients. The nodes are the variables while the edges are partial correlations between them. Green edges are positive and red are negative. Gray edges represent a point-biserial correlation between a continuous and a dichotomous variable. The thickness of the edge represents the strength of that correlation. The ring around each variable is the predictability value.

**Figure 6 diseases-12-00289-f006:**
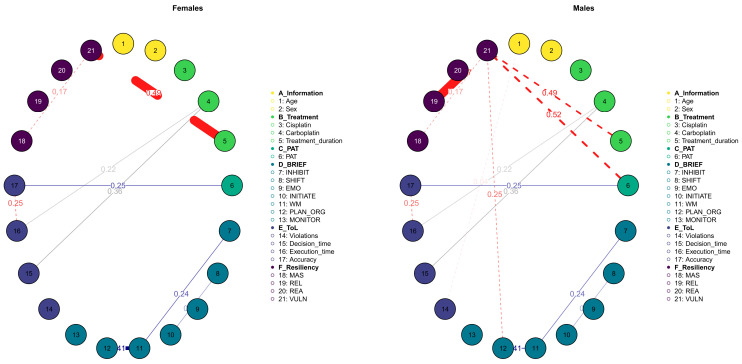
Moderated Gaussian Graphical Model. The two networks represent the same model but moderating the variable 2 (sex) in terms of 1 or 2 (females and males). The edges represent the conditional dependencies between variables. Red ones are negative and blue ones are positive.

**Figure 7 diseases-12-00289-f007:**
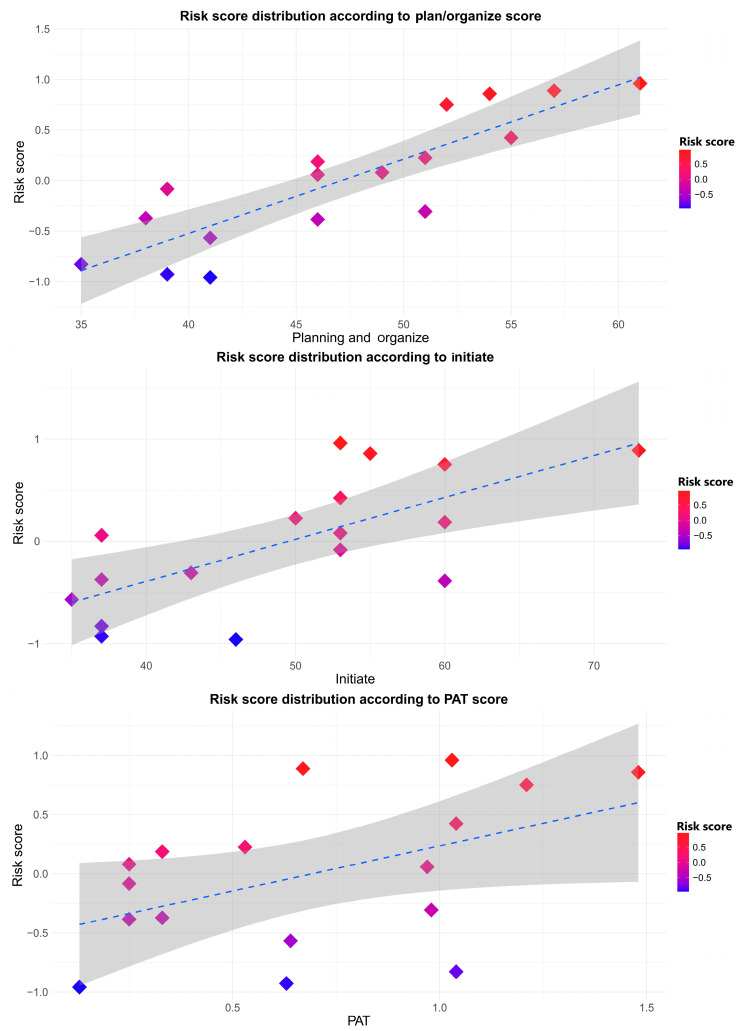
Cartesian representation of the three variables that had the most impact on the risk scores of the patients. Each dot represents a patient. The scores of the variables PAT, initiate, and plan/organize were impaired. The higher it was, the more negative it was.

**Table 1 diseases-12-00289-t001:** Centrality measures of each node of the network shown in Figure 5.

Node	Betweenness	Closeness	Strength
1	0.000	0.000	0.000
2	0.400	0.634	0.253
3	0.400	0.818	0.348
4	0.686	0.874	0.784
5	0.771	0.901	0.498
6	0.000	0.000	0.252
7	0.714	0.970	0.378
8	0.000	0.000	0.252
9	0.514	0.809	0.611
10	0.000	0.724	0.289
11	1.000	0.982	0.475
12	0.000	0.327	0.038
13	0.000	0.543	0.159
14	0.343	0.814	0.748
15	0.514	0.847	1.000
16	0.914	0.966	0.372
17	0.000	0.716	0.161
18	1.000	1.000	0.406
19	0.829	0.890	0.584

## Data Availability

The possibility of sharing raw data is only considered following a formal request to the corresponding author (Johanna M. C. Blom).

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
