# Peer review of "Exploring Sex-Based Neuropsychological Outcomes in Pediatric Brain Cancer Survivors: A Pilot Study"

_diseases, 2024, doi:10.3390/diseases12110289_

Round 1
Reviewer 1 Report
Comments and Suggestions for Authors
This study evaluated executive functioning and psychosocial factors, including resiliency, in children and adolescents treated with cisplatin and/or carboplatin. One of the major issues with this study was the very limited sample size as well as the very uneven number of females vs males comprising the sample. It should be noted that general lack of statistical significance may very well have been the result of the study having been underpowered due to the limited sample size. Of course, the sample size was reasonable given that this was a pilot study. The authors' inclusion criteria did limit participation to those who were treated with cisplatin and/or carboplatin. This is viewed as a positive since treatment variability in a limited sample would have prosed a problem and the authors do point out that these are standard treatments. Given that the neuropsychological literature supports the idea of a "chemobrain" following treatment for cancer, it may very well be that the cancer itself (depending on it's location) as well psychosocial variables and resilience all contribute to cognitive sequela. It is clear that the authors understand and appreciate the multiplicity of factors contributing to cognitive issues following treatment.
The neuropsychological tests and psychosocial measures used in this study were appropriate and are well known. It would have been nice if the authors would very briefly mention the validity and reliability of these measures. It may have been interesting if the authors had commented on their sample's performance relative to usual gender differences in executive functioning in persons of the same age range as the sample. In other words, it may be the case that the potential gender differences noted in the paper may be along the lines of what one might expect, independently of cancer treatment. In addition, it may have been useful if the authors had limited the age range of study participants because of age related differences in executive functioning. The authors were also forthright in discussing the limitations of this study as a guide to designing future studies. As the authors know, the family environment may have a profound impact on resiliency and a more robust assessment of resiliency may be helpful in future studies.
All in all, this was a well conceptualized pilot study which provided information that is likely to be useful in treatment planning and in designing future studies of the cognitive impact of chemotherapy treatment.
Reviewer 2 Report
Comments and Suggestions for Authors
The paper is interesting but I found it very confusing and very difficult to follow the rationale and identify the original contribution to the literature (perhaps, this comes through in the discussion). I would recommend that the authors make clear their study rationale and aims explicitly. I hope the below provides the authors with useful feedback.
1. The aims of the study seem to be to test the impact of cancer treatments upon cognitive functions. The results show no impact. The study goes on to report sex differences and a network analysis; however, the study does not provide a strong rationale for this question and it seems that the current study was not designed to test sex differences at study conception. Therefore, the validity of the methods and results is not clear. However, the discussion presents an interesting conclusion that may be interesting to the readership - cancer treatments may interact with sex. Whether this conclusion is valid, I suggest depends upon additional information needed.
2. The title does not seem to match the content of the paper.
3. Introduction: do you need to be clear about the survival rate of cancer in different countries? You quote information from Italy and America (why these two specifically?).
4. Rationale: the study rationale needs to be clearer – what new information is this study attempting to clarify? The authors provide numerous papers showing cognitive impact of cancer treatments.? What is this study adding to the literature base? One could argue that the current study is confusing, as it has reported no impact of cancer treatment yet the previous literature would suggest otherwise?.
5. Line 79 – 93: the discussion of mechanisms may be better placed in the discussion. Or earlier in the introduction. i.e. mechanistic discussion, followed by clinical research showing the cognitive impacts of cancer treatment etc.
6. Line 94 – 99: the literature on sex difference needs to be expanded and again, this study needs to present a clear understanding on what it is contributing to the literature. It appears that the study was not designed to test for sex differences and therefore, the rationale is not clear. If the authors do make this clearer, I would suggest it needs to be made explicit through out the paper and in the title.
6. Line100: states the aim of the study. It is not clear what is being added to current understanding. There is no mention of sex differences either.
7. Line 110-117: I would suggest this needs to be removed from the introduction.
Sample:
why 17 patients? Why only 6 males?
What do you consider adolescent – we would not consider a 22 year old to fit this category description?
The age range of diagnosis looks incorrect (line 126).
I am not sure how all participants demonstrated proficiency in language (i.e. 2 month old?) – line 127
More information about the participants in needed. I would suggest a table showing individual characteristics should be included (age, cancer, treatment etc., )
Methods:
Clarify who was giving their consent etc (patient or parent). Clarify the process for young children (e.g., 2 month old).
Tests: how were these administered to the young children? Can you provide confirmation of validity and reliability in this patient population etc.
Statistics:
Starts with test between male and female. This does not appear to be the primary aim of the study? Confirmation that the data fit assumptions is needed. Sex differences: how were other variables controlled or matched across sexes?
Line 159 – what normative data was used (line 159)? How was this established for the different age groups etc? more information is needed, with supporting references.
What is network analysis in this context? Provide reference. The analysis is very confusing. It is not clear why you have chosen to complete the analysis in this way, what research question are you addressing. You have also introduced a risk level – this is not clear?
Line 199: participant information doesn’t match that in the methods.
Line 200: used of the phrase ‘most participants’. What does this mean?
Line 249: is this conclusion valid. More information needed to explain the group analysis. The authors introduce gender for the first time too. This may need to be reviewed.
Is the paper trying to suggest that cancer treatment may impact the sexes differently? If so, a rationale for this needs to be provided and then tested
.
Comments on the Quality of English Languageno issues.
Reviewer 3 Report
Comments and Suggestions for Authors
Thank you for the opportunity to revise this work. The manuscript is well written, easy to read, the research design is good and the results clearly explained.
My only concern is about the correspondence between the sample size and the statistical methods employed in data analysis. For instance, a t-test is a parametric test that feels at home with sample sizes larger than 17 (it hopes for at least 30). It is not that the t-test is not robust enough to provide reliable results even with smaller sample sizes: it's about the stability/credibility of the findings, and the authors are aware of this.
The authors tried to prevent this sort of reproach by employing, for instance, cross-validation in their network analysis, and then building random forests to envelope the results into stable structures. But the main concern remains: 70% of the 17 patients means that the model was trained on 12-13 individuals, while classification models require hundreds of observations.
Now, as a human being I am not complaining that there are no more children with this issue so the sample size could be larger - that would be a crazy thing to hope for. As a scientist, however, I would need more evidence that the findings can be trusted. And I think that one solution could be that, when explaining their methodology, the authors could refer to previous papers: are there other authors employing the same methodology? Were those other papers using similar sample sizes? Is there any evidence in the extant literature that such a small sample size can lead to reliable results, or confirm previous findings resulted from larger sample sizes?
The authors already included sample size as an important limitation. However, with the right reference to previous research using similar methodologies maybe their argument can be strengthened.
Other than that, congratulations on a very good work.
Round 2
Reviewer 2 Report
Comments and Suggestions for Authors
The authors have clearly considered my feedback and have made changes to their paper to improve it's clarity.
Reviewer 3 Report
Comments and Suggestions for Authors
I thank the authors for the answers and explanations provided to my comments. I do believe that this is a good piece of research, and that the paper can be published.